# Fourier Series for Functions Related to Chebyshev Polynomials of the First Kind and Lucas Polynomials

**Taekyun Kim** [1,†], **Dae San Kim** [2,†] 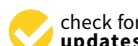, **Lee-Chae Jang** [3,*,†] **and Gwan-Woo Jang** [1,†]

[1]    Department of Mathematics, Kwangwoon University, Seoul 139-701, Korea; tkkim@kw.ac.kr (T.K.);
       gwjang@kw.ac.kr (G.-W.K.)
[2]    Department of Mathematics, Sogang University, Seoul 121-742, Korea; dskim@sogang.ac.kr
[3]    Graduate School of Education, Konkuk University, Seoul 139-701, Korea
[*]    Correspondence: lcjang@konkuk.ac.kr
[†]    These authors contributed equally to this work.

**Abstract:** In this paper, we derive Fourier series expansions for functions related to sums of finite products of Chebyshev polynomials of the first kind and of Lucas polynomials. From the Fourier series expansions, we are able to express those two kinds of sums of finite products of polynomials as linear combinations of Bernoulli polynomials.

**Keywords:** Fourier series; Chebyshev polynomials of the first kind; Lucas polynomials; Bernoulli polynomials

## 1. Introduction and Preliminaries

In this paper, we will consider some functions related to sums of finite products of Chebyshev polynomials of the first kind and of Lucas polynomials, and derive Fourier series expansions for them. Then, from the Fourier series expansions, we will be able to express those two kinds of sums of finite products of polynomials as linear combinations of Bernoulli polynomials.

Here, we would like to mention the following example as a motivation for studying these kinds of sums of finite products of special polynomials. Let us consider

$$\gamma_m(x) = \sum_{k=1}^{m-1} \frac{1}{k(m-k)} B_k(x) B_{m-k}(x), \quad (m \geq 2). \tag{1}$$

Then, in the same way as we will do in (14) and (17), it is possible to express $\gamma_m(x)$ in terms of Bernoulli polynomials by making use of the Fourier series expansion of $\gamma_m(<x>)$ (see (11)). Then, unlike the known involved proofs, from this expression, we can easily deduce the famous FPZ-identity (Faber-Pandharipande-Zagier identity) (see [1]) and a variant of the Miki's identity ([2–5]). Indeed, from the Fourier series expansion of $\gamma_m(<x>)$, we were able to deduce the following polynomial identity in (2), from which the variant of Miki's identity and FPZ-identity follow respectively by setting $x = 0$ and $x = \frac{1}{2}$ in the following:

$$
\begin{aligned}
&\sum_{k=1}^{m-1} \frac{1}{2k(2m-2k)} B_{2k}(x) B_{2m-2k}(x) + \frac{2}{2m-1} B_1(x) B_{2m-1}(x) \\
=&\frac{1}{m} \sum_{k=1}^{m} \frac{1}{2k} \binom{2m}{2k} B_{2k} B_{2m-2k}(x) + \frac{1}{m} H_{2m-1} B_{2m}(x) \\
&+ \frac{2}{2m-1} B_1(x) B_{2m-1}, \quad (m \geq 2),
\end{aligned}
\tag{2}
$$

where $H_m = \sum_{j=1}^{m} \frac{1}{j}$ are the harmonic numbers.

The reader refers to the Introduction of the paper [6] for some details on this.

Along the same line as the present paper, we obtained Fourier series expansions of sums of finite products of functions related to some Appell and some non-Appell polynomials and were able to express those sums of finite products of such polynomials in terms of Bernoulli polynomials as immediate corollaries. Indeed, they had been done for Appell polynomials like Bernoulli and Euler polynomials in [7,8], and, for quite a few non-Appell polynomials, namely Genocchi polynomials, Chebyshev polynomials of the second, third, fourth kinds, and Fibonacci, Legendre and Laguerre polynomials in [9–12]. Here, we let the reader refer to [13,14] as general references on orthogonal polynomials and to [15–17] as some recent papers on Lucas polynomials. As to some related results, we recommend the reader to look at the papers [7,8,12,18–22].

Chebyshev polynomials of the first kind have important applications in approximation theory. Indeed, their roots are used as nodes in polynomial interpolation and the resulting interpolation polynomial gives us a good polynomial approximation to a continuous function under the maximum norm. On the other hand, Lucas polynomials are useful in generating irreducible polynomials of high degree so that they have some applications in coding and cryptography. In addition, Lucas numbers are used in the areas relevant to operational research, statistics and computational mathematics, and allow us to find very large prime numbers in low complexity.

The Chebyshev polynomials $T_n(x)$ of the first kind and the Lucas polynomials $L_n(x)$ are respectively given by the recurrence relations as follows (see [13,14,16]):

$$T_{n+2}(x) = 2xT_{n+1}(x) - T_n(x), \quad (n \geq 0), \quad T_0(x) = 1, \quad T_1(x) = x, \tag{3}$$

$$L_{n+2}(x) = xL_{n+1}(x) + L_n(x), \quad (n \geq 0), \quad L_0(x) = 2, \quad L_1(x) = x. \tag{4}$$

From (3) and (4), we can easily derive the generating functions for $T_n(x)$ and $L_n(x)$ as follows:

$$F(t, x) = \frac{1 - xt}{1 - 2xt + t^2} = \sum_{n=0}^{\infty} T_n(x) t^n, \tag{5}$$

$$G(t, x) = \frac{2 - xt}{1 - xt - t^2} = \sum_{n=0}^{\infty} L_n(x) t^n. \tag{6}$$

The $T_n(x)$ and $L_n(x)$ are explicitly given as in the following:

$$T_n(x) = \frac{n}{2} \sum_{l=0}^{\left[\frac{n}{2}\right]} (-1)^l \frac{1}{n-l} \binom{n-l}{l} (2x)^{n-2l}, \quad (n \geq 1), \tag{7}$$

$$L_n(x) = n \sum_{l=0}^{\left[\frac{n}{2}\right]} \frac{1}{n-l} \binom{n-l}{l} x^{n-2l}, \quad (n \geq 1). \tag{8}$$

It is well known or easily checked from (7) and (8) that the two polynomials are related by

$$L_n(x) = 2i^{-n} T_n\left(\frac{ix}{2}\right), \quad i = \sqrt{-1}. \tag{9}$$

In terms of the generating function, the Bernoulli polynomials $B_n(x)$ are given by

$$\frac{t}{e^t - 1} e^{xt} = \sum_{n=0}^{\infty} B_n(x) \frac{t^n}{n!}. \tag{10}$$

For any real number $x$, the fractional part of $x$ is denoted by

$$< x >= x - [x] \in [0, 1),  \tag{11}$$

where $[x]$ indicates the greatest integer $\leq x$. For any integers $m, r$, with $m \geq 2, r \geq 1$, we let

$$
\begin{aligned}
\alpha_{m,r}(x) = &\sum_{l=0}^{m} \sum_{i_1+\cdots+i_{r+1}=m-l} \binom{r+l}{r} x^l T_{i_1}(x) \cdots T_{i_{r+1}}(x) \\
&- \sum_{l=0}^{m-2} \sum_{i_1+\cdots+i_{r+1}=m-l-2} \binom{r+l}{r} x^l T_{i_1}(x) \cdots T_{i_{r+1}}(x),
\end{aligned}  \tag{12}
$$

where the first and second inner sums run, respectively, over all nonnegative integers $i_1, \cdots, i_{r+1}$, with $i_1 + \cdots + i_{r+1} = m - l$, and with $i_1 + \cdots + i_{r+1} = m - l - 2$.

Then, we will consider the functions $\alpha_{m,r}(< x >)$, and derive their Fourier series expansions. From these Fourier series expansions, as a corollary, we can express $\alpha_{m,r}(x)$ as a linear combination of Bernoulli polynomials. Indeed, Theorems 1 and 2 are our results for the Fourier expansions of $\alpha_{m,r}(< x >)$, and Theorem 3 is those for the expressions of $\alpha_{m,r}(x)$ in terms of Bernoulli polynomials.

**Theorem 1.** *For any integers $m, r$ with $m \geq 2, r \geq 1$, we let*

$$\Delta_{m,r} = \frac{m+r}{r!} \sum_{l=0}^{\left[\frac{m-1}{2}\right]} \frac{(-1)^l}{m+r-l} \binom{m+r-l}{l} (m+r-2l)_r 2^{m-2l}.  \tag{13}$$

*Assume that $\Delta_{m,r} = 0$, for some integers $m, r$. Then, we have the following:*

*(a)*

$$
\begin{aligned}
&\sum_{l=0}^{m} \sum_{i_1+\cdots+i_{r+1}=m-l} \binom{r+l}{r} < x >^l T_{i_1}(< x >) \cdots T_{i_{r+1}}(< x >) \\
&- \sum_{l=0}^{m-2} \sum_{i_1+\cdots+i_{r+1}=m-l-2} \binom{r+l}{r} < x >^l T_{i_1}(< x >) \cdots T_{i_{r+1}}(< x >) \\
&= \frac{1}{2r} \Delta_{m+1,r-1} - \sum_{n=-\infty, n \neq 0}^{\infty} \left( \frac{1}{2r} \sum_{j=1}^{m} \frac{2^j (r+j-1)_j}{(2\pi i n)^j} \Delta_{m-j+1,r+j-1} \right) e^{2\pi i n x},
\end{aligned}
$$

*for all $x \in \mathbb{R}$. Here, the convergence is uniform.*

*(b)*

$$
\begin{aligned}
&\sum_{l=0}^{m} \sum_{i_1+\cdots+i_{r+1}=m-l} \binom{r+l}{r} < x >^l T_{i_1}(< x >) \cdots T_{i_{r+1}}(< x >) \\
&- \sum_{l=0}^{m-2} \sum_{i_1+\cdots+i_{r+1}=m-l-2} \binom{r+l}{r} < x >^l T_{i_1}(< x >) \cdots T_{i_{r+1}}(< x >) \\
&= \frac{1}{2r} \sum_{j=0, j \neq 1}^{m} 2^j \binom{r+j-1}{j} \Delta_{m-j+1,r+j-1} B_j(< x >),
\end{aligned}
$$

*for all $x \in \mathbb{R}$. Here, $(x)_r$ are the falling factorial polynomials defined by*

$$(x)_r = x(x-1) \cdots (x-r+1), \quad (r \geq 1), \quad (x)_0 = 1.$$

**Theorem 2.** *For any integers $m, r$ with $m \geq 2$, $r \geq 1$, let $\Delta_{m,r}$ be as in (13). Assume that $\Delta_{m,r} \neq 0$, for some positive integers $m, r$. Then, we have the following:*

*(a)*

$$\frac{1}{2r}\Delta_{m+1,r-1}$$

$$- \sum_{n=-\infty, n \neq 0}^{\infty} \left( \frac{1}{2r} \sum_{j=1}^{m} \frac{2^j (r+j-1)_j}{(2\pi i n)^j} \Delta_{m-j+1,r+j-1} \right) e^{2\pi i n x}$$

$$= \begin{cases} \sum_{l=0}^{m} \sum_{i_1+\cdots+i_{r+1}=m-l} \binom{r+l}{r} <x>^l T_{i_1}(<x>) \cdots T_{i_{r+1}}(<x>) \\ - \sum_{l=0}^{m-2} \sum_{i_1+\cdots+i_{r+1}=m-l-2} \binom{r+l}{r} <x>^l T_{i_1}(<x>) \cdots T_{i_{r+1}}(<x>), \\ \quad \text{for } x \in \mathbb{R} - \mathbb{Z}, \\ \frac{1}{2}\Delta_{m,r}, \quad \text{for } x \in \mathbb{Z}, \text{ and } m \text{ odd}, \\ (-1)^{\frac{m}{2}} \frac{m+r}{\frac{m}{2}+r} \binom{\frac{m}{2}+r}{r} + \frac{1}{2}\Delta_{m,r}, \quad \text{for } x \in \mathbb{Z}, \text{ and } m \text{ even}. \end{cases}$$

*(b)*

$$\frac{1}{2r} \sum_{j=0}^{m} 2^j \binom{r+j-1}{j} \Delta_{m-j+1,r+j-1} B_j(<x>)$$

$$= \sum_{l=0}^{m} \sum_{i_1+\cdots+i_{r+1}=m-l} \binom{r+l}{r} <x>^l T_{i_1}(<x>) \cdots T_{i_{r+1}}(<x>)$$

$$- \sum_{l=0}^{m-2} \sum_{i_1+\cdots+i_{r+1}=m-l-2} \binom{r+l}{r} <x>^l T_{i_1}(<x>) \cdots T_{i_{r+1}}(<x>),$$

$$\text{for all } x \in \mathbb{R} - \mathbb{Z};$$

$$\frac{1}{2r} \sum_{j=0, j \neq 1}^{m} 2^j \binom{r+j-1}{j} \Delta_{m-j+1,r+j-1} B_j(<x>)$$

$$= \begin{cases} \frac{1}{2r}\Delta_{m,r}, \quad \text{for } x \in \mathbb{Z}, \text{ and } m \text{ odd}, \\ (-1)^{\frac{m}{2}} \frac{m+r}{\frac{m}{2}+r} \binom{\frac{m}{2}+r}{r} + \frac{1}{2}\Delta_{m,r}, \quad \text{for } x \in \mathbb{Z}, \text{ and } m \text{ even}. \end{cases}$$

**Theorem 3.** *For any integers $m, r$ with $m \geq 2$, $r \geq 1$, we let $\Delta_{m,r}$ be as in (13). Then, we have the identity*

$$\sum_{l=0}^{m} \sum_{i_1+\cdots+i_{r+1}=m-l} \binom{r+l}{r} x^l T_{i_1}(x) \cdots T_{i_{r+1}}(x)$$

$$- \sum_{l=0}^{m-2} \sum_{i_1+\cdots+i_{r+1}=m-l-2} \binom{r+l}{r} x^l T_{i_1}(x) \cdots T_{i_{r+1}}(x) \tag{14}$$

$$= \frac{1}{2r} \sum_{j=0}^{m} 2^j \binom{r+j-1}{r-1} \Delta_{m-j+1,r+j-1} B_j(x).$$

In addition, for any integers $m, r$ with $m \geq 2$, $r \geq 1$, we put

$$\beta_{m,r}(x) = \sum_{l=0}^{m} \sum_{i_1+\cdots+i_{r+1}=m-l} \binom{r+l}{r} \left( \frac{x}{2} \right)^l L_{i_1}(x) \cdots L_{i_{r+1}}(x)$$

$$+ \sum_{l=0}^{m-2} \sum_{i_1+\cdots+i_{r+1}=m-l-2} \binom{r+l}{r} \left( \frac{x}{2} \right)^l L_{i_1}(x) \cdots L_{i_{r+1}}(x), \tag{15}$$

where the first and second inner sums are over all nonnegative integers $i_1, \cdots, i_{r+1}$, with $i_1 + \cdots + i_{r+1} = m - l$, and with $i_1 + \cdots + i_{r+1} = m - l - 2$, respectively.

Then, we will derive the Fourier series expansions of the functions $\beta_{m,r}(< x >)$, and express $\beta_{m,r}(x)$ in terms of Bernoulli polynomials, as an easy corollary to these Fourier series expansions.

In detail, Theorem 4 is our results for the Fourier series expansions of the functions $\beta_{m,r}(< x >)$, and Theorem 5 is those for the expressions of $\beta_{m,r}(x)$ in terms of Bernoulli polynomials.

**Theorem 4.** *For any integers $m, r$ with $m \geq 2, r \geq 1$, we let*

$$\Omega_{m,r} = \frac{2^{r+1}(m+r)}{r!} \sum_{l=0}^{\left[\frac{m-1}{2}\right]} \frac{1}{m+r-l} \binom{m+r-l}{l} (m+r-2l)_r. \tag{16}$$

*Then, we have the following:*

*(a)*

$$\frac{2}{r}\Omega_{m+1,r-1} - \sum_{n=-\infty, n\neq 0}^{\infty} \left( \frac{2}{r} \sum_{j=1}^{m} \left( \frac{1}{2\pi i n} \right)^j \frac{(r+j-1)_j}{2^j} \Omega_{m-j+1,r+j-1} \right) e^{2\pi i n x}$$

$$= \begin{cases} \sum_{l=0}^{m} \sum_{i_1+\cdots+i_{r+1}=m-l} \binom{r+l}{r} \left( \frac{<x>}{2} \right)^l L_{i_1}(< x >) \cdots L_{i_{r+1}}(< x >) \\ + \sum_{l=0}^{m-2} \sum_{i_1+\cdots+i_{r+1}=m-l-2} \binom{r+l}{r} \left( \frac{<x>}{2} \right)^l L_{i_1}(< x >) \cdots L_{i_{r+1}}(< x >), \\ \quad \text{for } x \in \mathbb{R} - \mathbb{Z}, \\ \frac{1}{2}\Omega_{m,r}, \quad \text{for } x \in \mathbb{Z}, \text{ and } m \text{ odd}, \\ \frac{1}{2}\Omega_{m,r} + 2^{r+1}\frac{m+r}{\frac{m}{2}+r}\binom{\frac{m}{2}+r}{r}, \quad \text{for } x \in \mathbb{Z}, \text{ and } m \text{ even}. \end{cases}$$

*(b)*

$$\frac{2}{r} \sum_{j=0}^{m} \binom{r+j-1}{j} \frac{1}{2^j} \Omega_{m-j+1,r+j-1} B_j(< x >)$$

$$= \sum_{l=0}^{m} \sum_{i_1+\cdots+i_{r+1}=m-l} \binom{r+l}{r} \left( \frac{<x>}{2} \right)^l L_{i_1}(< x >) \cdots L_{i_{r+1}}(< x >)$$

$$+ \sum_{l=0}^{m-2} \sum_{i_1+\cdots+i_{r+1}=m-l-2} \binom{r+l}{r} \left( \frac{<x>}{2} \right)^l L_{i_1}(< x >) \cdots L_{i_{r+1}}(< x >),$$

$$\text{for } x \in \mathbb{R} - \mathbb{Z};$$

$$\frac{2}{r} \sum_{j=0, j\neq 1}^{m} \binom{r+j-1}{j} \frac{1}{2^j} \Omega_{m-j+1,r+j-1} B_j(< x >)$$

$$= \begin{cases} \frac{1}{2}\Omega_{m,r}, \quad \text{for } x \in \mathbb{Z}, \text{ and } m \text{ odd}, \\ \frac{1}{2}\Omega_{m,r} + 2^{r+1}\frac{m+r}{\frac{m}{2}+r}\binom{\frac{m}{2}+r}{r}, \quad \text{for } x \in \mathbb{Z}, \text{ and } m \text{ even}. \end{cases}$$

**Theorem 5.** *For any integers $m, r$ with $m \geq 2$, $r \geq 1$, let $\Omega_{m,r}$ be as in* (16). *Then, we have the identity*

$$
\sum_{l=0}^{m} \sum_{i_1+\cdots+i_{r+1}=m-l} \binom{r+l}{r} \left(\frac{x}{2}\right)^l L_{i_1}(x) \cdots L_{i_{r+1}}(x)
$$

$$
+ \sum_{l=0}^{m-2} \sum_{i_1+\cdots+i_{r+1}=m-l-2} \binom{r+l}{r} \left(\frac{x}{2}\right)^l L_{i_1}(x) \cdots L_{i_{r+1}}(x) \tag{17}
$$

$$
= \frac{2}{r} \sum_{j=0}^{m} \binom{r+j-1}{r-1} \frac{1}{2^j} \Omega_{m-j+1,r+j-1} B_j(x).
$$

## 2. Fourier Series Expansions for Functions Related to the Chebyshev Polynomials of the First Kind

We will start with the next result, which plays a crucial role to our discussion in this section.

**Lemma 1.** *Let $m, r$ be integers with $m \geq 2$, $r \geq 1$. Then, we have the identity*

$$
\sum_{l=0}^{m} \sum_{i_1+\cdots+i_{r+1}=m-l} \binom{r+l}{r} x^l T_{i_1}(x) \cdots T_{i_{r+1}}(x)
$$

$$
- \sum_{l=0}^{m-2} \sum_{i_1+\cdots+i_{r+1}=m-l-2} \binom{r+l}{r} x^l T_{i_1}(x) \cdots T_{i_{r+1}}(x) \tag{18}
$$

$$
= \frac{1}{2^{r-1} r!} T_{m+r}^{(r)}(x),
$$

*where the first and second inner sums on the left-hand side are respectively over all nonnegative integers $i_1, \cdots, i_{r+1}$, with $i_1 + \cdots + i_{r+1} = m - l$, and with $i_1 + \cdots + i_{r+1} = m - l - 2$.*

**Proof.** By differentiating (5) $r$ times, we have

$$
\frac{\partial^r}{\partial x^r} F(t,x) = (t - t^3)(2t)^{r-1} r! (1 - 2xt + t^2)^{-(r+1)}, \quad (r \geq 1), \tag{19}
$$

$$
\frac{\partial^r}{\partial x^r} F(t,x) = \sum_{m=r}^{\infty} T_m^{(r)}(x) t^m = \sum_{m=0}^{\infty} T_{m+r}^{(r)}(x) t^{m+r}. \tag{20}
$$

Equations (19) and (20) give us

$$
\left(\frac{1}{1 - 2xt + t^2}\right)^{r+1} = \frac{1}{2^{r-1} r! (1 - t^2)} \sum_{m=0}^{\infty} T_{m+r}^{(r)}(x) t^m. \tag{21}
$$

On the other hand, using (5) and (21), we observe that

$$
\sum_{l=0}^{m} \sum_{i_1+\cdots+i_{r+1}=l} T_{i_1}(x) \cdots T_{i_{r+1}}(x) t^l
$$

$$
= \left(\sum_{l=0}^{\infty} T_l(x) t^l\right)^{r+1}
$$

$$
= (1 - xt)^{r+1} \left(\frac{1}{1 - 2xt + t^2}\right)^{r+1} \tag{22}
$$

$$
= (1 - xt)^{r+1} (1 - t^2)^{-1} \frac{1}{2^{r-1} r!} \sum_{m=0}^{\infty} T_{m+r}^{(r)}(x) t^m.
$$

From (22), we obtain

$$
\frac{1}{2^{r-1}r!} \sum_{m=0}^{\infty} T_{m+r}^{(r)}(x) t^m
$$

$$
= (1-t^2)(1-xt)^{-(r+1)} \sum_{l=0}^{\infty} \sum_{i_1+\cdots+i_{r+1}=l} T_{i_1}(x) \cdots T_{i_{r+1}}(x) t^l
$$

$$
= (1-t^2) \sum_{j=0}^{\infty} \binom{r+j}{j} x^j t^j \sum_{l=0}^{\infty} \sum_{i_1+\cdots+i_{r+1}=l} T_{i_1}(x) \cdots T_{i_{r+1}}(x) t^l
$$

$$
= \left( \sum_{j=0}^{\infty} \binom{r+j}{j} x^j t^j - \sum_{j=2}^{\infty} \binom{r+j-2}{j-2} x^{j-2} t^j \right)
$$

$$
\times \sum_{l=0}^{\infty} \sum_{i_1+\cdots+i_{r+1}=l} T_{i_1}(x) \cdots T_{i_{r+1}}(x)
$$

$$
= \sum_{m=0}^{\infty} \sum_{l=0}^{m} \binom{r+m-l}{m-l} x^{m-l} \sum_{i_1+\cdots+i_{r+1}=l} T_{i_1}(x) \cdots T_{i_{r+1}}(x) t^m
$$

$$
- \sum_{m=2}^{\infty} \sum_{l=0}^{m-2} \binom{r+m-l-2}{m-l-2} x^{m-l-2} \sum_{i_1+\cdots+i_{r+1}=l} T_{i_1}(x) \cdots T_{i_{r+1}}(x) t^m
$$

$$
= \sum_{m=0}^{\infty} \sum_{l=0}^{m} \binom{r+l}{l} x^l \sum_{i_1+\cdots+i_{r+1}=m-l} T_{i_1}(x) \cdots T_{i_{r+1}}(x) t^m
$$

$$
- \sum_{m=2}^{\infty} \sum_{l=0}^{m-2} \binom{r+l}{l} x^l \sum_{i_1+\cdots+i_{r+1}=m-l-2} T_{i_1}(x) \cdots T_{i_{r+1}}(x) t^m.
$$

(23)

By comparing both sides of (23) for $m \geq 2$, we get the desired result. $\square$

**Remark 1.** *Note that, from (23) with $m = 0, 1, 2$, we have*

$$
T_r^{(r)}(x) = 2^{r-1} r!,
$$

(24)

$$
T_{r+1}^{(r)}(x) = 2^r (r+1)! x,
$$

(25)

$$
T_{r+2}^{(r)}(x) = 2^{r-1}(r+2)! \left( 2x^2 - \frac{1}{r+1} \right).
$$

(26)

*From (7), we note that the $r$th derivative of $T_n(x)$ is given by*

$$
T_n^{(r)}(x) = \frac{n}{2} \sum_{l=0}^{\left[\frac{n-r}{2}\right]} (-1)^l \frac{1}{n-l} \binom{n-l}{l} 2^{n-2l} (n-2l)_r x^{n-2l-r}.
$$

(27)

*Then, combining (18) and (27), we obtain*

$$
\sum_{l=0}^{m} \sum_{i_1+\cdots+i_{r+1}=m-l} \binom{r+l}{l} x^l T_{i_1}(x) \cdots T_{i_{r+1}}(x)
$$

$$
- \sum_{l=0}^{m-2} \sum_{i_1+\cdots+i_{r+1}=m-l-2} \binom{r+l}{r} x^l T_{i_1}(x) \cdots T_{i_{r+1}}(x)
$$

$$
= \frac{2^m (m+r)}{r!} \sum_{l=0}^{\left[\frac{m}{2}\right]} \left( -\frac{1}{4} \right)^l \frac{1}{m+r-l} \binom{m+r-l}{l} (m+r-2l)_r x^{m-2l}.
$$

(28)

*As in* (12), *we let*

$$
\alpha_{m,r}(x) = \sum_{l=0}^{m} \sum_{i_1+\cdots+i_{r+1}=m-l} \binom{r+l}{l} x^l T_{i_1}(x) \cdots T_{i_{r+1}}(x)
$$

$$
- \sum_{l=0}^{m-2} \sum_{i_1+\cdots+i_{r+1}=m-l-2} \binom{r+l}{r} x^l T_{i_1}(x) \cdots T_{i_{r+1}}(x),
$$

(29)

*where* $m \geq 2$, *and* $r \geq 1$. *Now, we will consider the function*

$$
\alpha_{m,r}(<x>) = \sum_{l=0}^{m} \sum_{i_1+\cdots+i_{r+1}=m-l} \binom{r+l}{r} <x>^l T_{i_1}(<x>) \cdots T_{i_{r+1}}(<x>)
$$

$$
- \sum_{l=0}^{m-2} \sum_{i_1+\cdots+i_{r+1}=m-l-2} \binom{r+l}{r} <x>^l T_{i_1}(<x>) \cdots T_{i_{r+1}}(<x>),
$$

(30)

*which is defined on* $\mathbb{R}$ *and periodic with period* 1.

*The Fourier series of* $\alpha_{m,r}(<x>)$ *is*

$$
\sum_{n=-\infty}^{\infty} A_n^{(m,r)} e^{2\pi i n x},
$$

(31)

*where*

$$
A_n^{(m,r)} = \int_0^1 \alpha_{m,r}(<x>) e^{-2\pi i n x} dx
$$

$$
= \int_0^1 \alpha_{m,r}(x) e^{-2\pi i n x} dx.
$$

(32)

*For* $m \geq 2, r \geq 1$, *let us put*

$$
\Delta_{m,r} = \alpha_{m,r}(1) - \alpha_{m,r}(0),
$$

(33)

*where we note that*

$$
\alpha_{m,r}(0) = \begin{cases} 0, & \text{if } m \text{ is odd,} \\ (-1)^{\frac{m}{2}} \frac{m+r}{\frac{m}{2}+r} \binom{\frac{m}{2}+r}{r}, & \text{if } m \text{ is even.} \end{cases}
$$

(34)

*From* (28), (33) *and* (34), *we obtain*

$$
\Delta_{m,r} = \frac{m+r}{r!} \sum_{l=0}^{\left[\frac{m-1}{2}\right]} \frac{(-1)^l}{m+r-l} \binom{m+r-l}{l} (m+r-2l)_r 2^{m-2l}.
$$

(35)

*It is immediate to see from* (18) *that*

$$
\frac{d}{dx} \alpha_{m,r}(x) = \frac{1}{2^{r-1} r!} T_{m+r}^{(r+1)}(x)
$$

$$
= 2(r+1) \alpha_{m-1,r+1}(x).
$$

(36)

*In turn,* (36) *yields the following:*

$$
\frac{d}{dx} \left( \frac{\alpha_{m+1,r-1}(x)}{2r} \right) = \alpha_{m,r}(x),
$$

(37)

$$\int_0^1 \alpha_{m,r}(x)dx = \frac{1}{2r}\Delta_{m+1,r-1},$$

(38)

$$\alpha_{m,r}(0) = \alpha_{m,r}(1) \Longleftrightarrow \Delta_{m,r} = 0.$$

(39)

We are now going to determine the Fourier coefficients $A_n^{(m,r)}$.

Case 1: $n \neq 0$.

$$\begin{aligned}
A_n^{(m,r)} &= \int_0^1 \alpha_{m,r}(x)e^{-2\pi inx}dx \\
&= -\frac{1}{2\pi in}\left[\alpha_{m,r}(x)e^{-2\pi inx}\right]_0^1 + \frac{1}{2\pi in}\int_0^1 \left(\frac{d}{dx}\alpha_{m,r}(x)\right)e^{-2\pi inx}dx \\
&= \frac{2(r+1)}{2\pi in}\int_0^1 \alpha_{m-1,r+1}(x)e^{-2\pi inx}dx - \frac{1}{2\pi in}\Delta_{m,r} \\
&= \frac{2(r+1)}{2\pi in}A_n^{(m-1,r+1)} - \frac{1}{2\pi in}\Delta_{m,r}.
\end{aligned}$$

Thus, we have shown the following recursive relation:

$$A_n^{(m,r)} = \frac{2(r+1)}{2\pi in}A_n^{(m-1,r+1)} - \frac{1}{2\pi in}\Delta_{m,r},$$

(40)

which in turn gives the following expression

$$A_n^{(m,r)} = -\frac{1}{2r}\sum_{j=1}^m \frac{2^j(r+j-1)_j}{(2\pi in)^j}\Delta_{m-j+1,r+j-1}.$$

(41)

Case 2: $n = 0$.

$$A_0^{(m,r)} = \int_0^1 \alpha_{m,r}(x)dx = \frac{1}{2r}\Delta_{m+1,r-1}.$$

(42)

To proceed further, we recall the following facts about Bernoulli function:

(a)   for $m \geq 2$,

$$B_m(< x >) = -m!\sum_{n=-\infty,n\neq 0}^{\infty} \frac{e^{2\pi inx}}{(2\pi in)^m},$$

(43)

(b)   for $m = 1$,

$$-\sum_{n=-\infty,n\neq 0}^{\infty} \frac{e^{2\pi inx}}{2\pi in} = \begin{cases} B_1(< x >), & \text{for } x \in \mathbb{R} - \mathbb{Z}, \\ 0, & \text{for } x \in \mathbb{Z}. \end{cases}$$

(44)

*From* (41)–(44), *we get the next Fourier series expansion of* $\alpha_{m,r}(<x>)$

$$
\frac{1}{2r}\Delta_{m+1,r-1} - \sum_{n=-\infty,n\neq 0}^{\infty}\left(\frac{1}{2r}\sum_{j=1}^{m}\frac{2^j(r+j-1)_j}{(2\pi in)^j}\Delta_{m-j+1,r+j-1}\right)e^{2\pi inx}
$$

$$
= \frac{1}{2r}\Delta_{m+1,r-1} + \frac{1}{2r}\sum_{j=1}^{m}2^j\binom{r+j-1}{j}\Delta_{m-j+1,r+j-1}\left(-j!\sum_{n=-\infty,n\neq 0}^{\infty}\frac{e^{2\pi inx}}{(2\pi in)^j}\right)
$$

$$
= \frac{1}{2r}\sum_{j=0,j\neq 1}^{m}2^j\binom{r+j-1}{j}\Delta_{m-j+1,r+j-1}B_j(<x>) \tag{45}
$$

$$
+ \Delta_{m,r}\times\begin{cases}B_1(<x>), & \text{for } x\in\mathbb{R}-\mathbb{Z},\\ 0, & \text{for } x\in\mathbb{Z}.\end{cases}
$$

*Evidently, the function* $\alpha_{m,r}(<x>)$, $(m\geq 2, r\geq 1)$ *is piecewise* $C^\infty$. *Moreover,* $\alpha_{m,r}(<x>)$ *is continuous for those integers* $m, r$ *with* $\Delta_{m,r}=0$, *and discontinuous with jump discontinuities at integers for those integers* $m, r$ *with* $\Delta_{m,r}\neq 0$. *Hence, for* $\Delta_{m,r}=0$, *the Fourier series of* $\alpha_{m,r}(<x>)$ *converges uniformly to* $\alpha_{m,r}(<x>)$; *for* $\Delta_{m,r}\neq 0$, *the Fourier series of* $\alpha_{m,r}(<x>)$ *converges pointwise to* $\alpha_{m,r}(<x>)$, *for* $x\in\mathbb{R}-\mathbb{Z}$, *and converges to*

$$
\frac{1}{2}(\alpha_{m,r}(0)+\alpha_{m,r}(1)) = \alpha_{m,r}(0)+\frac{1}{2}\Delta_{m,r}, \tag{46}
$$

*for* $x\in\mathbb{Z}$. *Now, from* (45), (46), *and these observations, we have Theorems* 1 *and* 2 *in Section* 1. *We remark here that Theorem* 3 *in Section* 1 *follows immediately from* (b) *of Theorems* 1 *and* 2. *Before closing this section, we will illustrate the identity* (14), *for* $m=2, r=1$ *and also for* $m=3, r=1$. *For this, we first note that*

$$
T_0(x)=1, T_1(x)=x, T_2(x)=2x^2-1, T_3(x)=4x^3-3x, \tag{47}
$$

$$
B_0(x)=1, B_1(x)=x-\frac{1}{2}, B_2(x)=x^2-x+\frac{1}{6}, B_3(x)=x^3-\frac{3}{2}x^2+\frac{1}{2}x. \tag{48}
$$

*By* (35), *we have*

$$
\Delta_{3,0}=2, \Delta_{2,1}=12, \Delta_{1,2}=6, \tag{49}
$$

$$
\Delta_{4,0}=0, \Delta_{3,1}=16, \Delta_{2,2}=24, \Delta_{1,3}=8. \tag{50}
$$

*In addition, from* (47), *we see that*

$$
\sum_{i+j=3}T_i(x)T_j(x)=12x^3-8x, \quad \sum_{i+j=2}T_i(x)T_j(x)=5x^2-2,
$$

$$
\sum_{i+j=1}T_i(x)T_j(x)=2x, \quad \sum_{i+j=0}T_i(x)T_j(x)=1. \tag{51}
$$

*Now, we see from* (48)–(51) *that the identity in* (14) *for* $m=2, r=1$ *and that, for* $m=3, r=1$ *correspond respectively to*

$$
\sum_{i+j=2}T_i(x)T_j(x)+2x\sum_{i+j=1}T_i(x)T_j(x)+3x^2\sum_{i+j=0}T_i(x)T_j(x)-\sum_{i+j=0}T_i(x)T_j(x)
$$

$$
= B_0(x)+12B_1(x)+12B_2(x)=12x^2-3,
$$

$$\sum_{i+j=3} T_i(x)T_j(x) + 2x \sum_{i+j=2} T_i(x)T_j(x) + 3x^2 \sum_{i+j=1} T_i(x)T_j(x)$$

$$+ 4x^3 \sum_{i+j=0} T_i(x)T_j(x) - \sum_{i+j=1} T_i(x)T_j(x) - 2x \sum_{i+j=0} T_i(x)T_j(x)$$

$$= 16B_1(x) + 48B_2(x) + 32B_3(x) = 32x^3 - 16x.$$

## 3. Fourier Series Expansions for Functions Related to the Lucas Polynomials

The proof for the next lemma will be omitted, as this can be shown just as in the case of Lemma 1.

**Lemma 2.** *Let $m, r$ be integers with $m \geq 2$, $r \geq 1$. The following identity holds true:*

$$\sum_{l=0}^{m} \sum_{i_1+\cdots+i_{r+1}=m-l} \binom{r+l}{r} \left(\frac{x}{2}\right)^l L_{i_1}(x) \cdots L_{i_{r+1}}(x)$$

$$+ \sum_{l=0}^{m-2} \sum_{i_1+\cdots+i_{r+1}=m-l-2} \binom{r+l}{r} \left(\frac{x}{2}\right)^l L_{i_1}(x) \cdots L_{i_{r+1}}(x) \tag{52}$$

$$= \frac{2^{r+1}}{r!} L_{m+r}^{(r)}(x),$$

*where the first and second inner sums on the left-hand side are respectively over all nonnegative integers $i_1, \cdots, i_{r+1}$, with $i_1 + \cdots + i_{r+1} = m - l$, and with $i_1 + \cdots + i_{r+1} = m - l - 2$.*

**Remark 2.** *The identity in* (52) *follows from*

$$\frac{2^{r+1}}{r!} \sum_{m=0}^{\infty} L_{m+r}^{(r)}(x) t^m$$

$$= \sum_{m=0}^{\infty} \sum_{l=0}^{m} \binom{r+l}{r} \left(\frac{x}{2}\right)^l \sum_{i_1+\cdots+i_{r+1}=m-l} L_{i_1}(x) \cdots L_{i_{r+1}}(x) t^m \tag{53}$$

$$+ \sum_{m=2}^{\infty} \sum_{l=0}^{m-2} \binom{r+l}{r} \left(\frac{x}{2}\right)^l \sum_{i_1+\cdots+i_{r+1}=m-l-2} L_{i_1}(x) \cdots L_{i_{r+1}}(x) t^m.$$

*With $m = 0, 1, 2$ in* (53)*, we obtain*

$$L_r^{(r)}(x) = r!, \tag{54}$$

$$L_{r+1}^{(r)}(x) = (r+1)! x, \tag{55}$$

$$L_{r+2}^{(r)}(x) = \frac{(r+1)!}{2} \left\{ \left(r+1+\frac{1}{2^r}\right) x^2 + \frac{1}{2^{r-1}} \right\}. \tag{56}$$

*We see from* (8) *that the rth derivative of $L_n(x)$ is given by*

$$L_n^{(r)}(x) = n \sum_{l=0}^{\left[\frac{n-r}{2}\right]} \frac{1}{n-l} \binom{n-l}{l} (n-2l)_r x^{n-2l-r}. \tag{57}$$

*Then, combining (52) and (57), we have*

$$
\sum_{l=0}^{m} \sum_{i_1+\cdots+i_{r+1}=m-l} \binom{r+l}{r} \left(\frac{x}{2}\right)^l L_{i_1}(x) \cdots L_{i_{r+1}}(x)
$$
$$
+ \sum_{l=0}^{m-2} \sum_{i_1+\cdots+i_{r+1}=m-l-2} \binom{r+l}{r} \left(\frac{x}{2}\right)^l L_{i_1}(x) \cdots L_{i_{r+1}}(x) \tag{58}
$$
$$
= \frac{2^{r+1}(m+r)}{r!} \sum_{l=0}^{\left[\frac{m}{2}\right]} \frac{1}{m+r-l} \binom{m+r-l}{l}(m+r-2l)_r x^{m-2l}.
$$

*For $m \geq 2$, and $r \geq 1$, as in (15), we let*

$$
\beta_{m,r}(x) = \sum_{l=0}^{m} \sum_{i_1+\cdots+i_{r+1}=m-l} \binom{r+l}{r} \left(\frac{x}{2}\right)^l L_{i_1}(x) \cdots L_{i_{r+1}}(x)
$$
$$
+ \sum_{l=0}^{m-2} \sum_{i_1+\cdots+i_{r+1}=m-l-2} \binom{r+l}{r} \left(\frac{x}{2}\right)^l L_{i_1}(x) \cdots L_{i_{r+1}}(x). \tag{59}
$$

*Now, we will consider the function*

$$
\beta_{m,r}(<x>) = \sum_{l=0}^{m} \sum_{i_1+\cdots+i_{r+1}=m-l} \binom{r+l}{r} \left(\frac{<x>}{2}\right)^l L_{i_1}(<x>) \cdots L_{i_{r+1}}(<x>)
$$
$$
+ \sum_{l=0}^{m-2} \sum_{i_1+\cdots+i_{r+1}=m-l-2} \binom{r+l}{r} \left(\frac{<x>}{2}\right)^l L_{i_1}(<x>) \cdots L_{i_{r+1}}(<x>), \tag{60}
$$

*which is defined on $\mathbb{R}$ and periodic with period 1. The Fourier series of $\beta_{m,r}(<x>)$ is*

$$
\sum_{n=-\infty}^{\infty} B_n^{(m,r)} e^{2\pi i n x}, \tag{61}
$$

*where*

$$
B_n^{(m,r)} = \int_0^1 \beta_{m,r}(<x>) e^{-2\pi i n x} dx
$$
$$
= \int_0^1 \beta_{m,r}(x) e^{-2\pi i n x} dx. \tag{62}
$$

*For $m \geq 2$, and $r \geq 1$, we put*

$$
\Omega_{m,r} = \beta_{m,r}(1) - \beta_{m,r}(0). \tag{63}
$$

*Then, from (58) and (63), we see that*

$$
\Omega_{m,r} = \frac{2^{r+1}(m+r)}{r!} \sum_{l=0}^{\left[\frac{m-1}{2}\right]} \frac{1}{m+r-l} \binom{m+r-l}{l}(m+r-2l)_r, \tag{64}
$$

*where we observe that*

$$
\beta_{m,r}(0) = \begin{cases} 0, & \text{if } m \text{ is odd,} \\ 2^{r+1} \frac{m+r}{\frac{m}{2}+r} \binom{\frac{m}{2}+r}{r}, & \text{if } m \text{ is even.} \end{cases} \tag{65}
$$

*The following can be easily derived from* (52):

$$\frac{d}{dx}\beta_{m,r}(x) = \frac{2^{r+1}}{r!}L_{m+r}^{(r+1)}(x) = \left(\frac{r+1}{2}\right)\beta_{m-1,r+1}(x), \tag{66}$$

$$\frac{d}{dx}\left(\frac{2}{r}\beta_{m+1,r-1}(x)\right) = \beta_{m,r}(x), \tag{67}$$

$$\int_0^1 \beta_{m,r}(x)dx = \frac{2}{r}\Omega_{m+1,r-1}, \tag{68}$$

$$\beta_{m,r}(0) = \beta_{m,r}(1) \iff \Omega_{m,r} = 0. \tag{69}$$

*We are now ready to determine the Fourier coefficients* $B_n^{(m,r)}$.

*Case 1:* $n \neq 0$.

$$
\begin{aligned}
B_n^{(m,r)} &= \int_0^1 \beta_{m,r}(x)e^{-2\pi inx}dx \\
&= -\frac{1}{2\pi in}\left[\beta_{m,r}(x)e^{-2\pi inx}\right]_0^1 + \frac{1}{2\pi in}\int_0^1\left(\frac{d}{dx}\beta_{m,r}(x)\right)e^{-2\pi inx}dx \\
&= \frac{1}{2\pi in}\frac{r+1}{2}\int_0^1 \beta_{m-1,r+1}(x)e^{-2\pi inx}dx - \frac{1}{2\pi in}\Omega_{m,r} \\
&= \frac{1}{2\pi in}\frac{r+1}{2}B_n^{(m-1,r+1)} - \frac{1}{2\pi in}\Omega_{m,r}.
\end{aligned}
\tag{70}
$$

*Thus, we have derived the following recurrence relation:*

$$B_n^{(m,r)} = \frac{1}{2\pi in}\frac{r+1}{2}B_n^{(m-1,r+1)} - \frac{1}{2\pi in}\Omega_{m,r}, \tag{71}$$

*from which we readily have*

$$B_n^{(m,r)} = -\frac{2}{r}\sum_{j=1}^m \left(\frac{1}{2\pi in}\right)^j \frac{(r+j-1)_j}{2^j}\Omega_{m-j+1,r+j-1}. \tag{72}$$

*Case 2:* $n = 0$.

$$B_0^{(m,r)} = \int_0^1 \beta_{m,r}(x)dx = \frac{2}{r}\Omega_{m+1,r-1}. \tag{73}$$

*Then, from* (72)*,* (73)*,* (43)*, and* (44)*, we obtain the following Fourier series expansion of* $\beta_{m,r}(<x>)$*, which is given by*

$$
\begin{aligned}
&\frac{2}{r}\Omega_{m+1,r-1} - \sum_{n=-\infty,n\neq 0}^{\infty}\left(\frac{2}{r}\sum_{j=1}^m\left(\frac{1}{2\pi in}\right)^j\frac{(r+j-1)_j}{2^j}\Omega_{m-j+1,r+j-1}\right)e^{2\pi inx} \\
&= \frac{2}{r}\Omega_{m+1,r-1} + \frac{2}{r}\sum_{j=1}^m\binom{r+j-1}{j}\frac{1}{2^j}\Omega_{m-j+1,r+j-1}\left(-j!\sum_{n=-\infty,n\neq 0}^{\infty}\frac{e^{2\pi inx}}{(2\pi in)^j}\right) \\
&= \frac{2}{r}\sum_{j=0,j\neq 1}^m\binom{r+j-1}{j}\frac{1}{2^j}\Omega_{m-j+1,r+j-1}B_j(<x>) \\
&\quad + \Omega_{m,r} \times \begin{cases} B_1(<x>), & \text{for } x \in \mathbb{R} - \mathbb{Z}, \\ 0, & \text{for } x \in \mathbb{Z}. \end{cases}
\end{aligned}
\tag{74}
$$

Note here that $\Omega_{m,r} > 0$, for any $m \geq 2$, $r \geq 1$. Thus, $\beta_{m,r}(< x >)$ is piecewise $C^\infty$, and discontinuous with jump discontinuities at integers. Thus, the Fourier series of $\beta_{m,r}(< x >)$ converges pointwise to $\beta_{m,r}(< x >)$, for $x \in \mathbb{R} - \mathbb{Z}$, and converges to

$$\frac{1}{2}(\beta_{m,r}(0) + \beta_{m,r}(1)) = \beta_{m,r}(0) + \frac{1}{2}\Omega_{m,r} \tag{75}$$

for $x \in \mathbb{Z}$.

This observation together with (74) and (75) yields Theorem 4 in Section 1. Here, we observe that Theorem 5 in Section 1 follows from (b) of Theorem 4.

From (9), we can easily deduce that

$$2^{r+1}i^{-m}\alpha_{m,r}\left(\frac{ix}{2}\right) = \beta_{m,r}(x). \tag{76}$$

In turn, by Theorems 3 and 5, (76) yields the following theorem.

**Theorem 6.** *For any integers $m, r$ with $m \geq 2$, $r \geq 1$, we let*

$$\Omega_{m,r} = \frac{2^{r+1}(m+r)}{r!}\sum_{l=0}^{\left[\frac{m-1}{2}\right]}\frac{1}{m+r-l}\binom{m+r-l}{l}(m+r-2l)_r.$$

*Let $m, r$ be integers with $m \geq 2$, $r \geq 1$. Then, we have the following identity:*

$$2^{r-1}\sum_{j=0}^{m}2^j\binom{r+j-1}{j}\Delta_{m-j+1,r+j-1}B_j\left(\frac{ix}{2}\right)$$

$$= i^m\sum_{j=0}^{m}\binom{r+j-1}{j}\frac{1}{2^j}\Omega_{m-j+1,r+j-1}B_j(x),$$

*where $\Delta_{m,r}$ and $\Omega_{m,r}$ are respectively as in (35) and (64).*

## 4. Conclusions

In general, the connection problem is concerned with determining the coefficients $c_{nm}(k)$ in the representation of the product of two polynomials $r_n(x)$ and $s_m(x)$ as linear combinations of an arbitrary polynomial sequence $\{p_k(x)\}_{k\geq 0}$:

$$r_n(x)s_m(x) = \sum_{k=0}^{n+m}c_{nm}(k)p_k(x). \tag{77}$$

As a generalization of this and motivated by the example in (1), we considered the problem of representing sums of finite products of Chebyshev polynomials of the first kind and those of Lucas polynomials in terms of Bernoulli polynomials. We accomplished this by deriving the Fourier series expansions of the functions related to those two kinds of sums of finite products of polynomials. Finally, we remark here that it is certainly possible to represent such sums of finite products of polynomials by some orthogonal polynomials, which is our ongoing project.

**Author Contributions:** T.K. and D.S.K. conceived the framework and structured the whole paper; D.S.K. wrote the paper; L.-C.J. and G.-W.J. checked the results of the paper; D.S.K. and T.K. completed the revision of the article.

**Acknowledgments:** We would like to thank the referees for their comments and suggestions which improved the original manuscript in its present form.

**Conflicts of Interest:** The authors declare no conflict of interest.

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
