# Peer review of "Fourier Series for Functions Related to Chebyshev Polynomials of the First Kind and Lucas Polynomials"

_mathematics, doi:10.3390/math6120276_

Round 1
Reviewer 1 Report
The paper is just a sequence of theorem, no introduction, no discussion.
There is a section "Preliminaries", but it is not enough.
This definitely has to be changed.
In the Abstract, the authors say that "This can be viewed as solving a
classical connection problem". And then in the paper
"We note here that our results are solutions for specific connection problems.
However, the following example gives us another motivation for studying these
kinds of sums of finite products of special polynomials."
So, this is the only time the authors mention "connection problems", so, the abstract also should be changed,
since the authors do not discuss connections problems, but actually talk about another motivation for their study.
I recommend the authors to connect their theorems and proofs (the proofs mostly look ok) together,
otherwise it is almost a dot-point list of the results.
Author Response
Point-by-point response to referee
Reviewer1.
The paper is just a sequence of theorem, no introduction, no discussion.
There is a section "Preliminaries", but it is not enough.
This definitely has to be changed.
(response) I moved the next to the last paragraph in Section 1 to the second paragraph of Section 1. Also, I added two paragraphs, one as the first paragraph and the other as the third one in Section 1.
I think that the referee’s suggestion is reasonable, since the original manuscript begins and proceeds with formulas, and introduction and discussion appear only at the very end of Section 1.
In the Abstract, the authors say that "This can be viewed as solving a
classical connection problem". And then in the paper
"We note here that our results are solutions for specific connection problems.
However, the following example gives us another motivation for studying these
kinds of sums of finite products of special polynomials."
So, this is the only time the authors mention "connection problems", so, the abstract also should be changed, since the authors do not discuss connections problems, but actually talk about another motivation for their study.
(response) Following the referee’s suggestion, I removed phrases “connection problems” from the Abstract and the paragraph at the end of Section 1.
I recommend the authors to connect their theorems and proofs (the proofs mostly look ok) together, otherwise it is almost a dot-point list of the results.
(response) As the proofs are very lengthy, I think it is not a good idea to connect theorems and proofs. We would rather stick to the present form.

Reviewer 2 Report
This paper presents an interesting result of series involving products of classical orthogonal polynomials. These include products of Chebyshev and Lucas polynomials. This referee has checked sufficiently many entries in the paper and is convinced that the results are correct.
There is one comment that I believe will make the paper more accessible to a larger audience. In the introduction the authors should include the statements of Theorems 2.2 and 2.3 (containing the Fourier expansions) and then refer to them in the proofs. In this form, the reader will find all the results in one place. Also, since many readers might have problems "digesting" these formulas, it would be very useful to spell in detail an example of each of these theorems, with small values of the parameters. This will help appreciate the value of the general theorems.
Author Response
Point-by-point response to referee
Reviewer2.
This paper presents an interesting result of series involving products of
classical orthogonal polynomials. These include products of Chebyshev and
Lucas polynomials. This referee has checked sufficiently many entries in the
paper and is convinced that the results are correct.
There is one comment that I believe will make the paper more accessible to a
larger audience. In the introduction the authors should include the statements
of Theorems 2.2 and 2.3 (containing the Fourier expansions) and then refer
to them in the proofs. In this form, the reader will find all the results in one
place. Also, since many readers might have problems "digesting" these
formulas, it would be very useful to spell in detail an example of each of these
theorems, with small values of the parameters. This will help appreciate the
value of the general theorems.
(response)
All the results in this paper are now stated in in Section 1, as the referee directed.
I added an example for the sums of finite products of Chebyshev polynomials of the first kind at the very end of Section 1.
But I didn’t give that for the sums of finite products of Lucas polynomials.
This is because I think that the interested reader can easily produce such examples and feel that this paper is already somewhat lengthy.
Reviewer 3 Report
This is a really nice paper, clearly-written paper filled with elegant identities and techniques.
I believe the readers of this journal would find it both interesting and useful. Based on a thorough reading of the paper, I recommend this paper for publication in Mathematics.
Page 8, line 90, "relation." should be "relation"
Page 8, line 91, "expression." should be "expression"
Page 9, line 96, "(<x>)." should be "(<x>)"
Page 9, line 106, "following." should be "following:"
Page 10, line 113, "lowing." should be "lowing:"
Page 12, line 122, "true." should be "true:"
Page 14, line 140, "(3.1)." should be "(3.1):"
Page 14, line 146, "relation." should be "relation"
Page 15, line 156, "following." should be "following:"
Page 16, line 161, "identity." should be "identity:"
Page 16, line 161, "B_j(x)." should be "B_j(x),"
Author Response
Point-by-point response to referee
Reviewer2.
This is a really nice paper, clearly-written paper filled with elegant identities and techniques.
I believe the readers of this journal would find it both interesting and useful. Based on a thorough reading of the paper, I recommend this paper for publication in Mathematics.
Page 8, line 90, "relation." should be "relation"
Page 8, line 91, "expression." should be "expression"
Page 9, line 96, "(<x>)." should be "(<x>)"
Page 9, line 106, "following." should be "following:"
Page 10, line 113, "lowing." should be "lowing:"
Page 12, line 122, "true." should be "true:"
Page 14, line 140, "(3.1)." should be "(3.1):"
Page 14, line 146, "relation." should be "relation"
Page 15, line 156, "following." should be "following:"
Page 16, line 161, "identity." should be "identity:"
Page 16, line 161, "B_j(x)." should be "B_j(x),"
(response) All of the above suggestions are changed as the referee directed.

Round 2
Reviewer 1 Report
The paper was improved. Now there is an introduction. There is still a lot of room for improvement, since this is a collection of formula substitutions (no motivation and where it leads, the advantage of the new results, etc.). All these makes it hard to assess the significance.
Author Response
Point-by-point response to referee
Reviewer1.
for Authors
The paper was improved. Now there is an introduction. There is still a lot of room for improvement, since this is a collection of formula substitutions (no motivation and where it leads, the advantage of the new results, etc.). All these makes it hard to assess the significance.
(response)
(1) To emphasize the motivation, I moved a slightly expanded version of the last paragraph in Section 1 to the second one, which is the following:
Here we would like to mention the following example as a motivation for studying these kinds of sums of finite products of special polynomials. Let us consider
\begin{equation}\begin{split}\label{a}
\gamma_m(x) = \sum_{k=1}^{m-1}
\frac{1}{k(m-k)}B_k(x)B_{m-k}(x), \;\;(m\geq 2).
\end{split}\end{equation}
Then, in the same way as we will do in \eqref{12} and \eqref{15}, it is possible to express
$\gamma_m(x)$ in terms of Bernoulli polynomials by making use of the Fourier
series expansion of $\gamma_m(<x>)$ (see \eqref{09}). Then, unlike the known involved proofs, from this expression we can easily deduce the famous FPZ-identity (Faber-Pandharipande-Zagier identity) (see [5]) and a variant of
the Miki's identity ([4,6,17,18]). Indeed, from the Fourier series expansion of $\gamma_m(<x>)$ we were able to deduce the following polynomial identity in \eqref{b}, from which the variant of Miki's identity and FPZ-identity follow respectively by setting $x=0$ and $x=\frac{1}{2}$ in the following:
\begin{equation}\begin{split}\label{b}
& \sum_{k=1}^{m-1}\frac{1}{2k\left(2m-2k\right)}B_{2k}\left(x\right)B_{2m-2k}\left(x\right)+\frac{2}{2m-1}B_{1}\left(x\right)B_{2m-1}\left(x\right)\\
= & \frac{1}{m}\sum_{k=1}^{m}\frac{1}{2k}\dbinom{2m}{2k}B_{2k}B_{2m-2k}\left(x\right)+\frac{1}{m}H_{2m-1}B_{2m}\left(x\right)\\
&+\frac{2}{2m-1}B_{1}\left(x\right)B_{2m-1},\quad\left(m\ge2\right),
\end{split}\end{equation}
where $H_{m}=\sum_{j=1}^{m}\frac{1}{j}$ are the harmonic numbers.
The reader refers to the Introduction of the paper [15] for some details on this.
(2) Also, I added the ‘Conclusion’ section.
(3) Upon Reviewer 2’s request, all theorems are stated in Section 1, and at the very end of Section 2 some examples of our result are given.